# Prospective Analysis of *TERT* Promoter Mutations in Papillary Thyroid Carcinoma at a Single Institution

**DOI:** 10.3390/jcm10102179

**Published:** 2021-05-18

**Authors:** Yun-Suk Choi, Seong-Woon Choi, Jin-Wook Yi

**Affiliations:** Department of Surgery, Inha University Hospital & College of Medicine, Inchoen 22332, Korea; yunsukki@gmail.com (Y.-S.C.); unchoi87@gmail.com (S.-W.C.)

**Keywords:** PTC, telomere, *TERT* promoter mutation, *BRAF*^V600E^ mutation

## Abstract

Background: Papillary thyroid cancer (PTC) has the highest cancer incidence in Korea. It is known that some thyroid cancers have aggressive clinical behavior and a poor prognosis. Genomic studies have described some somatic mutations that are related to the aggressive features of thyroid cancer, such as the *BRAF*^V600E^ mutation. Recently, *TERT* promoter mutations were identified and reported as poor prognostic factors in PTC. Our aim was to identify the frequency and clinical impact of *TERT* promoter mutation in PTC. Methods: Analysis of both *BRAF*^V600E^ and *TERT* promoter mutations in thyroidectomy specimens began in February 2019. As of December 2020, 622 patients had been tested. Data were prospectively collected and retrospectively reviewed to ascertain clinical and pathologic variables. Results: *TERT* promoter mutations were identified in 13 patients (2.09%); 12 had the C228T mutation, and one had the C216T mutation. In total, ten patients had the *BRAF*^V600E^ mutation. *TERT* promoter mutation was significantly associated with advanced age (46.795 ± 12.616 versus 65.692 ± 13.628 years, *p* < 0.001), large tumor size (1.006 ± 0.829 versus 2.285 ± 1.938 cm, *p* = 0.035), extrathyroidal extension, surgical margin involvement, angioinvasion, *BRAF*^V600E^ mutation and advanced TNM stage, a higher MACIS score and a high proportion of radioactive iodine therapy application. Logistic regression showed that lymphatic and angioinvasion and *BRAF*^V600E^ mutation were predictive of *TERT* promoter mutation. Conclusions: Our study is the first to report the prospective results of *TERT* promoter mutations at a single tertiary hospital in Incheon, Korea. PTC with *TERT* promoter mutation was associated with more aggressive behavior than PTC with wild-type *TERT* gene status.

## 1. Introduction

Thyroid cancer (TC) is one of the most prevalent cancers in Korea and frequently occurs in adolescents and young adults between 15 and 34 years of age [1]. According to the 2019 national statistics, 52,070 new TC patients and 2170 deaths were expected to occur in the United States, and 11,667 new patients and 340 deaths were expected to occur in the Republic of Korea [1,2]. Thyroid cancers are classified as differentiated thyroid cancer (DTC) including papillary thyroid carcinoma (PTC) and follicular thyroid carcinoma (FTC); poorly differentiated carcinoma (PDTC) and anaplastic carcinoma (AC); medullary thyroid carcinoma (MTC); and lymphoma. Among them, differentiated cancer is reported to account for 98.3% of all thyroid cancers, and 79.3% are papillary thyroid carcinoma (PTC) [3]. Treatment principle of thyroid cancer is the surgical resection of thyroid gland with regional lymph node dissection, and optional radioactive iodine therapy.

Patients with undifferentiated thyroid cancer have been reported to have a poor prognosis, including those with poorly differentiated thyroid cancer (PDTC) and anaplastic thyroid cancer (ATC) [3]. However, some patients with DTC also have a poor prognosis [4]. Known prognosis-related factors in PTC are large tumor size, extrathyroidal extension, and lymph node metastasis based on the AJCC staging system [5]. Genomic studies have demonstrated the clinical association of the *BRAF*^V600E^ mutation with poor prognostic factors in PTC [6]. However, a recent study showed that the prevalence of *BRAF* mutation is very high, above 80% in Korea, and the clinical significance of *BRAF*^V600E^ mutation has faded for most PTCs [7,8].

Recently, it has been shown that promoter mutations in telomerase reverse transcriptase (*TERT)* gene are associated with a poor prognosis in thyroid cancer, especially with concurrent *BRAF*^V600E^ mutation [9]. *TERT* is the catalytic protein subunit of telomerase, a ribonucleoprotein complex that plays a key role in cellular immortality by maintaining telomere length at the end of chromosomes [10]. In cancer cells, the *TERT* gene may act as an oncogene by promoting cell immortality via the telomere-independent proliferation of cancer cells [11]. In TC, three major *TERT* promoter mutations have been reported, namely, chr5:1, 295 228 C > T(C228T) and chr5:1, 295 250 C > T(C250T), which represent nucleotide changes of −124 C > T and −146 C > T from the ATG translation start codon in the promoter site of the *TERT* gene [12]. Since 2019, our center has started the *TERT* promoter mutation test in surgically resected thyroid cancer tissue. Our aim was, therefore, to evaluate the frequency of TERT promoter mutations of PTC and its clinical meaning in Korean PTC patients at a single institution.

## 2. Materials and Methods

### 2.1. Patient Data

From February 2019 to December 2020, we analyzed 622 thyroid cancer patients who were tested and reported the full result for the *BRAF*^V600E^ and *TERT* promoter mutations in their surgically resected thyroid cancer tissue at Inha University Hospital, Incheon, Korea. The clinical significance and estimated cost of the *BRAF* and *TERT* promoter mutation tests were sufficiently explained to the patients, and the tests were performed only in the patients who agreed to the tests. *BRAF*^V600E^ and *TERT* promoter mutation tests were only performed for patients with histologically confirmed PTC; for patients with other cancer types, such as follicular thyroid cancer and medullary thyroid cancer, the *BRAF* and *TERT* mutation test was not performed.

All surgeries were performed by a single endocrine surgeon (JW Yi). Data were prospectively collected and retrospectively reviewed to ascertain clinical and pathologic variables. According to the *TERT* promoter mutation status, we divided the patients into two groups: *TERT* wild-type and *TERT* promoter mutation. Clinical variables included patient age at diagnosis, sex, preoperative fine needle aspiration cytology (Bethesda category), tumor location, extent of thyroidectomy and lymph node dissection, surgical approach, conventional open and endoscopic/robotic approach, operation time and estimated blood loss during surgery. Variables associated with radioactive iodine therapy (RAI) were serum unstimulated thyroglobulin (Tg) at 3 months postoperatively (ng/mL), RAI dose (mCi), stimulated thyroid stimulation hormone (TSH) level before the RAI (µIU/mL) and TSH-stimulated Tg level (ng/mL).

Pathologic variables included the histologic subtypes for PTC, largest tumor diameter, tumor multiplicity, extrathyroidal extension either microscopically or grossly, surgical margin status, and lymphatic and angioinvasion. Staging of T, N, and overall stage was estimated according to the AJCC 8th edition of differentiated thyroid cancer [13]. The MACIS score was calculated by two independent surgeons as previously reported [14].

### 2.2. Mutation Test for the BRAF ^V600E^ and TERT Promoter Mutation

The *BRAF*^V600E^ mutation test was performed by the Department of Pathology in our hospital. Genomic DNA was extracted from formalin-fixed paraffin-embedded (FFPE) tissue using the QIAamp DSP DNA FFPE Tissue Kit (QUIGEN®, Hilden, Germany). Our hospital used the PNAClamp^TM^
*BRAF* Mutation Detection Kit (PANAGENE Inc., Daejeon, Korea) for the *BRAF^V600E^* mutation test.

*TERT* promoter mutation testing was performed in the Department of Pathology, Seoul St. Mary’s Hospital, Korea. Amplification of *TERT* promoter was done by nested polymerase chain reaction (PCR). Initially, 235-bp PCR amplicon was amplified using forward 5′-AGTGGATTCGCGGGCACAGA-3′ and reverse 5′-CAGCGCTGCCTGAAACTC-3′ primers. Second, 163-bp PCR amplicon was amplified using forward 5′-GTCCTGCCCCTTCACCTT-3′ and reverse 5′-CAGCGCTGCCTGAAACTC-3′ primers. Bidirectional Sanger sequencing in both directions was done using primers that used in the second PCR [7]. 

### 2.3. Statistical Analysis and Ethics

For the statistical analysis, we used R programming language version 3.6.1 [15]. Chi-square or Fisher’s exact test was applied to the cross-table analysis according to sample size. For the mean comparison, unpaired *t*-test was used. To identify the variables associated with *TERT* promoter mutation status, logistic regression with the backward selection method was applied. Statistical significance was defined as a *p*-value under 0.05.

Ethical approval for this study was obtained from the institutional review board of our hospital (IRB number: 2021-03-037 (Approval date: 29 March 2021)).

## 3. Results

*TERT* promotor mutation was detected in 13 (2.09%) of the 622 patients. Table 1 presents the details of patients who had *TERT* promoter mutations. The C228T mutation was detected in 12 patients, and the C216T mutation was detected in one patient. *BRAF*^V600E^ mutation was detected in 11 patients. The mean age of the patients was 65.692 ± 13.628 years, and 12 were female. Fine needle aspiration cytology for 12 patients revealed suspicious malignant and malignant disease, but one patient had benign disease. Twelve patients underwent total thyroidectomy with central node lymph node dissection. Modified radical neck dissection (MRND) was performed in three patients. The PTC subtypes were as follows: 9 classic PTCs, 2 tall cell variants, and one solid and follicular variant. Specifically, one patient (patient 7) showed lateral neck node recurrence within 1 year, and the same *TERT* C228T mutation was found in the metastatic lymph nodes. Patient 10 was considered to have a benign nodule 7.7 cm before the surgery, but follicular variant PTC was diagnosed.

Table 2 describes the clinical and pathologic characteristics according to *TERT* promoter mutation status. Compared to the wild-type *TERT* promoter group, the *TERT* promoter mutation group was significantly older in age (65.692 ± 13.628 versus 46.795 ± 12.616, *p* < 0.001), with eleven patients over 55 years of age. Regarding the surgical aspects, total thyroidectomy was preferred, and the frequency of MRND was significantly higher in the *TERT* promoter mutation group. Histologically, the variant type of PTC was more common in the TERT promoter mutation group. The tumor size was larger (1.006 ± 0.829 versus 2.285 ± 1.938, *p* = 0.035), and extrathyroidal extension, resection margin involvement and angioinvasion were significantly higher in the *TERT* promoter mutation group, and the T, N, and overall stage of these patients were more advanced. The MACIS score was also higher in the *TERT* promoter mutation group. Radioactive iodine therapy was also performed at a significantly higher frequency in the *TERT* promoter mutation group. Sex, preoperative fine needle aspiration cytology, tumor location, surgical approach, surgery time, blood loss and lymphatic invasion were not significantly different between the two groups.

Table 3 shows the variables according to radioactive iodine therapy. Compared to the wild-type *TERT* promoter group, the *TERT* promoter mutation group showed a higher postoperative 3-month thyroglobulin (Tg) level (0.593 ± 2.578 versus 46.528 ± 158.241, *p* = 0.336), and the proportion of Tg levels ≥1 was significantly higher. The stimulated Tg level before the initial RAI was also higher in the *TERT* promoter mutation group (8.961 ± 52.728 versus 54.594 ± 145.762, *p* = 0.349). Although the proportion of stimulated Tg levels ≥1 was higher in the *TERT* promoter mutation group, this difference did not reach statistical significance.

Factors associated with *TERT* promotor mutation by logistic regression analysis are presented in Table 4. In the univariable analysis, age over 55, variant-type PTC, tumor size over 1 cm, the presence of extrathyroidal extension (either microscopic or gross), surgical margin involvement, angioinvasion, T3, T4 stage, Nb stage, overall disease Stages II and III, and a higher MACIS risk score were significantly associated with *TERT* promoter mutation. In the multivariable analysis, age over 55 years, lymphatic invasion, angioinvasion and *BRAF*^V600E^ mutation were considered reliable variables for *TERT* promoter mutation status.

## 4. Discussion

Although most TC have a good prognosis, some TC exhibit unusually aggressive behavior and are associated with a poor prognosis. To distinguish more aggressive from less aggressive thyroid cancer, several biomarkers have been widely studied. When the *BRAF*^V600E^ mutation was first being studied, it was reported as a predictor of clinical aggressiveness and poor prognosis in thyroid cancer [16]. Over time, the detection rate of *BRAF*^V600E^ mutation in papillary thyroid cancer has been very high, up to 74% to 80% in previous literature [17,18]. In our study, the prevalence of *BRAF*^V600E^ mutation in PTC was 82.6%. As such, the current *BRAF*^V600E^ mutation is being used to increase the diagnostic sensitivity of fine aspiration cytology or needle biopsy specimens rather than as a prognostic factor [19].

Teloermase is an enzyme necessary to preserve telomere maintenance. Cancer cells produce telomerase and preserve the length of the teletomere to prevent apoptosis of cancer cells and allow unlimited proliferation of cells. In thyroid cancer cells, driver mutation of the *BRAF* gene activates the mitogen-activated protein kinase pathway, which induces overexpression of the E-twenty-six (ETS) transcription factor. ETS can bind to *TERT* promoter mutation sites such as C228T or C250T, which increases exonic *TERT* gene overexpression. Overexpression of the *TERT* gene then increases tumor cell telomere stability and is linked to cell overdevelopment and malignant transformation. These molecular consequences contribute to cancer cell immortality, which is thought to be associated with the aggressive behavior and poor survival of PTC [20,21].

Several studies have reported the clinical significance of *TERT* promoter mutation. In the meta-analysis conducted by Moon et al., 8.3% of patients exhibited concurrent *BRAF*^V600E^ and *TERT* promoter mutations, which showed an association with advanced stage, extrathyroidal extension, lymph node metastasis, distant metastasis, increased risk of recurrence and worse survival [9]. In another study reported by Vuong et al., the rate of *TERT* promoter mutation with *BRAF*^V600E^ mutation was 5.8% and has been associated with aggressive behaviors of cancer, such as advanced T stage and overall stage, extrathyroidal extension, lymph node metastasis, distant metastasis and increased recurrence [22]. According to the results of previous studies, although the incidence of *TERT* promoter mutation is reported to be low, it may be worthwhile to test *TERT* promoter mutation in thyroid cancer patients because they are associated with a poor clinical prognosis.

In South Korea, the *TERT* promoter mutation test in surgically removed TC tissue was approved by the Ministry of Health and Welfare and implemented in December 2018. Since then, many hospitals in Korea have begun performing *TERT* promoter mutation tests for thyroid cancer patients. In 2020, Kim et al. was the first to report 1-year *TERT* promoter mutation test results in thyroid cancer patients [7]. In this study, the detection rate of *TERT* promoter mutation was 2.8% (20/724); C228T mutation was found in 1.9% (14/724), and C250T mutation was identified in 0.3% (2/724). A novel C216T mutation was identified in 0.6% (4/724). The incidence of *TERT* promoter mutation in their cohort was lower than that in previous studies (5.8% to 8.3%, respectively) [9,22]. *TERT* promoter mutation was associated with older age, large tumor size, extrathyroidal extension, advanced T and N stage, and higher recurrence rates according to the American Thyroid Association recurrence risk estimator.

Our hospital started testing for *TERT* promoter mutation in PTC in February 2019. Due to the high cost of each gene mutation test, approximately 150 USD, *BRAF* and *TERT* mutation tests were performed only for patients who agreed to the test prior to surgery. The benefit and cost of the *BRAF* gene and *TERT* gene mutation tests were fully explained to the patient. According to our results, the incidence of *TERT* promoter mutation was 2.1% (13 of 622), in which 12 patients carried the C228T mutation and one patient carried the C216T mutation. C250T mutation was not identified in our cohort. The incidence of TERT promotor mutation in Korean cohort was about 2%, which is less frequent than previous studies about 10% of TERT promotor mutation [23]. This is probably caused by racial differences but further research is needed.

In our analysis, the presence of *TERT* promoter mutation was associated with older age, total thyroidectomy, lateral lymph node metastasis, variant-type PTC, large tumor size, the presence of extrathyroidal extension, positive surgical margin involvement, blood vessel invasion, *BRAF*^V600E^ mutation, advanced T and N stages, advanced overall stage and higher MACIS score, as described in Table 2. Based on the logistic regression analysis results shown in Table 4, lymphatic invasion, angioinvasion, *BRAF*^V600E^ mutation and advanced T stage were predictive variables for *TERT* promoter mutation. Among the patients in our study cohort, one patient presented early recurrence on the lateral neck node within the one-year follow-up, and the same *TERT* promoter mutation in C228T was found in her lateral node after surgery [4]. This finding suggested that *TERT* promoter mutation in thyroid cancer can also affect aggressive metastatic neck lymph nodes, and short-term follow-up is required for patients who have *TERT* promoter mutation. One patient (Patient Number 10) underwent right lobectomy due to the large goiter for her benign nodule, and the final pathology revealed a 7.7 cm follicular variant PTC. According to this case, follicular variant PTC can also harbor the *TERT* promoter mutation without the *BRAF*^V600E^ mutation. The clinical meaning of *TERT* promoter mutation in follicular variant PTC or follicular thyroid cancer is still unknown, and a large number of cases should be accumulated. One patient carried the C216T mutation (Patient number 3). Of note, the impact of *TERT* C216T is still unknown. As the same mutation was also found in other studies, long-term follow-up is needed to determine what the clinical significance of this mutation [7].

Our findings suggest that *TERT* promoter mutation is associated with aggressive clinical features of PTC. However, as the proportion of *TERT* promoter mutation is very low, it is necessary to think about their clinical use, and cost-benefit analysis should be considered. Therefore, based on the results of our analysis, we suggest that it is helpful to identify candidate thyroid cancer patients with a poor prognosis to selectively perform the *TERT* gene mutation test in more advanced patients rather than in all patients. We also identified the *TERT* promoter mutation in three papillary microcarcinoma patients who received total thyroidectomy. From a clinical point of view, when a *TERT* promoter mutation is detected after thyroid lobectomy in a thyroid micropapillary cancer patient, whether to remove the opposite thyroid gland and perform radioiodine treatment or follow-up without surgery and RAI has not yet been established. We think that more prospective research is needed in this area.

The limitation of this study is the relatively short-term follow-up and the inability to assess recurrence according to the *TERT* promoter mutation status. A second limitation is that *TERT* promoter mutation testing was not performed in our hospital directly. The cancer tissue was prepared in a paraffin block and sent to another hospital, and this process may affect the detection of *TERT* promoter mutation. In the future, if our hospital can establish a setting to perform *TERT* promoter mutation testing, it is expected that more accurate results will be obtained.

In conclusion, we report a large cohort study in which *TERT* promoter mutation status was investigated in Korean patients who underwent TC surgery. *TERT* promoter mutation was found to be a significantly worse prognostic factor in TC. However, due to the cost-benefit aspect of this testing, it should be considered for selective patients of an older age with aggressive features of PTC.

## Figures and Tables

**Table 1 jcm-10-02179-t001:** Clinical and pathologic variables of the patients with *TERT* promoter mutation.

Patient Number	Age	Sex	Bethesda Category	Thyroidectomy	Node Dissection	Variant	Tumor Size (cm)	Extrathyroidal Extension	Lymphatic Invasion	Vascular Invasion	*BRAF* ^V600E^ Mutation	*TERT* Promoter Mutation
1	61	Female	Malignant (VI)	Total	CND	Classic	1.1	Strap muscle	Absent	Absent	Present	C228T
2	48	Male	Malignant (VI)	Total	CND	Classic	0.5	Absent	Indeterminate	Indeterminate	Present	C228T
3	69	Female	Malignant (VI)	Total	CND	Classic	0.6	Absent	Absent	Absent	Present	C216T
4	63	Female	Malignant (VI)	Total	CND	Solid	2.5	Strap muscle	Indeterminate	Present	None	C228T
5	65	Female	Malignant (VI)	Total	CND	Classic	0.6	Absent	Absent	Absent	Present	C228T
6	78	Female	Malignant (VI)	Total	CND	Classic	3.5	Trachea, esophagus	Present	Absent	Present	C228T
7	68	Female	Malignant (VI)	Total	MRND	Classic	2.5	Strap muscle	Present	Absent	Present	C228T
8	34	Female	Malignant (VI)	Total	MRND	Classic	1.6	Absent	Present	Present	Present	C228T
9	70	Female	Suspicious malignant (V)	Total	MRND	Tall cell	3.5	Trachea	Present	Present	Present	C228T
10	89	Female	Benign (II)	Lobectomy	None	Follicular	7.7	Absent	Absent	Absent	None	C228T
11	78	Female	Malignant (VI)	Total	CND	Tall cell	2.2	Strap muscle	Absent	Absent	Present	C228T
12	65	Female	Malignant (VI)	Total	CND	Classic	2.5	Trachea	Present	Absent	Present	C228T
13	66	Female	Malignant (VI)	Total	CND	Classic	0.9	Trachea	Absent	Absent	Present	C228T

**Table 2 jcm-10-02179-t002:** Clinical and pathologic characteristics of the patients according to *TERT* promoter mutation status.

Variables	All (*n* = 622)	Wild Type *TERT* (*n* = 609)	*TERT* Promoter Mutation (*n* = 13)	*p*-Value
Age (years, mean ± sd)	47.190 ± 12.912	46.795 ± 12.616	65.692 ± 13.628	<0.001
<55	438	436	2	<0.001
≥55	184	173	11	
Sex				
Male	132	131	1	0.319
Female	490	478	12	
Bethesda category				
I	4	4	0	0.869
II	23	22	1	
III	30	30	0	
IV	14	14	0	
V	123	121	2	
VI	428	418	10	
Tumor location				
Right	94	266	5	0.821
Left	13	239	5	
Isthmus	244	13	0	
Bilateral	271	91	3	
Thyroidectomy				
Lobectomy, isthmectomy	251	250	1	0.019
Total, completion	371	359	12	
Lymph node dissection				
Less than central	574	569	9	0.013
Lateral node dissection	48	44	4	
Surgical approach				
Open	260	252	8	0.163
Endoscopic, robotic	362	357	5	
Operation time (minutes, mean ± sd)	125.660 ± 49.471	125.218 ± 49.043	146.154 ± 65.770	0.276
Blood loss (ml, mean ± sd)	51.892 ± 73.573	51.900 ± 73.371	51.538 ± 85.814	0.988
Histologic subtypes				
PTC (Classic type)	583	574	9	0.006
Variant	39	35	4	
Tumor size (cm, mean ± sd)	1.033 ± 0.883	1.006 ± 0.829	2.285 ± 1.938	0.035
≤1 cm	415	411	4	0.013
>1 cm	207	198	9	
Multiplicity				
Single	372	366	6	0.393
Multiple	250	243	7	
Extrathyroidal extension				
Absent	456	451	5	0.008
Present	166	158	8	
Surgical margin				
Negative	552	545	7	0.001
Positive	70	64	6	
Lymphatic invasion				
Absent	426	420	6	0.195
Indeterminate	68	66	2	
Present	128	123	5	
Angioinvasion				
Absent	547	538	9	<0.001
Indeterminate	63	62	1	
Present	12	9	3	
*BRAF*^V600E^ mutation				
Negative	108	106	2	<0.001
Positive	514	503	11	
T stage				
T1	499	495	4	<0.001
T2	26	26	0	
T3	66	61	5	
T4	31	27	4	
N stage				
N0, Nx	362	354	8	0.003
N1a	212	211	1	
N1b	48	44	4	
Overall stage				
I	544	540	4	<0.001
II	62	57	5	
III	16	12	4	
IV	0	0	0	
MACIS score				
<6	557	555	4	<0.001
6–6.69	57	52	5	
≥7	6	2	4	
Radioactive iodine therapy				
No	394	391	3	0.006
Yes	228	218	10	

**Table 3 jcm-10-02179-t003:** Radioactive iodine therapy-related variables (*n* = 228).

Variables	Wild Type *TERT* (*n* = 218)	*TERT* Promoter Mutation (*n* = 10)	*p*-Value
Postoperative 3 months Tg * (mean ± SD, ng/mL)	0.593 ± 2.578	46.528 ± 158.241	0.336
<1 (Number of patients)	197	6	0.015
≥1 (Number of patients)	21	4	
1st RAI ^†^ dose (mean ± SD, mCi)	91.972 ± 37.518	98.000 ± 43.919	0.678
TSH level before RAI (mean ± SD, µIU/mL)	126.322 ± 46.325	158.982 ± 77.127	0.216
Stimulated Tg level before RAI (mean ± SD, ng/mL)	8.961 ± 52.728	54.594 ± 145.762	0.349
<1 (Number of patients)	114	3	0.206
≥1 (Number of patients)	104	7	

* Tg thyroglobulin. ^†^ radioactive iodine therapy.

**Table 4 jcm-10-02179-t004:** Logistic regression analysis according to *TERT* promoter mutation status.

Variables (Reference)		Univariable		Multivariable	
		Odds Ratio	*p* Value	Odds Ratio (95% CI)	*p* Value
Age (<55 years)	≥55 years	13.861	0.001	18.673	0.03
Sex (Male)	Female	3.289	0.255		
Pathologic subtype (Classic)	Variant	7.289	0.001		
Multiplicity (Single)	Multiple	1.757	0.316		
Tumor size (≤1 cm)	>1 cm	4.67	0.011		
Extrathyroidal extension (Absent)	Present	4.567	0.009		
Margin (Negative)	Positive	7.299	0.001		
Lymphatic invasion (Absent)	Indeterminate	2.121	0.363	204.417	0.034
	Present	2.846	0.089	1.773	0.573
Angioinvasion (Absent)	Indeterminate	0.964	0.973	0.060	0.276
	Present	19.926	<0.001	124.638	0.004
*BRAF* (Wild type)	V600E mutation	1.159	0.849	1030.344	0.022
T stage (T1)	T2	0	0.991		
	T3	10.143	0.001		
	T4	18.333	<0.001		
N stage (N0, Nx)	N1a	0.21	0.142	0.000	0.991
	N1b	4.023	0.028	5.582	0.117
AJCC stage (I)	II	11.842	<0.001		
	III	45	<0.001		
MACIS risk score (<6)	6–6.69	13.341	<0.001	8.309	0.324
	≥7	277.5	<0.001		0.99

Logistic regression model with backward selection.

## Data Availability

Data sharing not applicable. No new data were created or analyzed in this study. Data sharing is not applicable to this article.

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
