# Peer review of "Prospective Analysis of TERT Promoter Mutations in Papillary Thyroid Carcinoma at a Single Institution"

_jcm, 2021, doi:10.3390/jcm10102179_

Round 1
Reviewer 1 Report
Thank you for the opportunity to review the manuscript titled "Prospective Analysis of TERT Promoter Mutations in Papillary Thyroid Carcinoma at a Single Institution."
This manuscript describes the prevalence of TERT promoter mutations in 622 surgically-resected PTC samples at Inha University Hospital. BRAF p.V600E mutation status was also assessed.
In general, this was a well written manuscript that described the clinical and pathological details of all PTC samples studied. Although this type of study is not novel, it helps boost available data from multiple institutions to be studied as a whole. However, this manuscript lacks good discussion on the significance of the findings.
I have a few comments and suggestions:
- TYPOGRAPHICAL ERROR. Introduction, line 55. There are excessive spaces and lack of commas when describing the genomic location of the TERT 228 and 250 known hotspot locations.
- TYPOGRAPHICAL ERROR. Methods, section 2.1 Patient data mentions that “From February 2019 to December 2021…” Please correct, as Dec 2021 is in the future.
- Methods for BRAF p.V600E and TERT promoter mutation detections should be more thorough. Specifically, vendor details for the kit listed for BRAF detection shows this method by qPCR which is a very sensitive method for mutation detection. However, TERT promoter mutation analysis was done by Sanger sequencing, which has an analytical sensitivity of roughly 20%. Please elaborate on the mutation detection methods in the Methods section. Different analytical sensitives of the methods used for BRAF and TERT mutation detection may yield false negative results for one of the assays.
- Additionally, specific to Sanger sequencing of TERT promoter region – this is a very GC-rich region. Did Sanger sequencing work and generate a result, either positive or negative, for 100% of the 622 sample cohort?
- A meta-analysis performed by Liu et. al. (PMID 27833153) shows the average prevalence of TERT promoter mutations in PTC to be ~10%. Your study showed approx. 2% prevalence. What may contribute to very low TERT mutation positive prevalence?
- Since the significance of TERT C216T is unknown, and is acknowledged in the manuscript as such, please describe why it was included in the group of samples with identified TERT C228T mutations. The manuscript is written such that identification of a TERT promoter mutation correlates with more aggressive PTC behavior. Since C216T mutation has unknown significance, please elaborate on inclusion of the sample with this variant with all other TERT C228T positive cases, as there is no data to support its association with aggressiveness?
- The aim of this paper was “to evaluate the frequency of TERT promoter mutations in consecutive cases of PTC and its clinical meaning…” Prevalence data and clinical features of the tumors and disease at diagnosis are presented in this paper. Please elaborate on the clinical meaning. This is where the manuscript is really lacking merit. Is there any follow up data for the 622 sample cohort? There was mention of only one follow-up in line 222. There are several mentions of the association of TERT promoter mutations with poor prognosis. Is that what was found in this patient set? Did any patients present with distant metastasis? Were there BRAF-positive, TERT-negative samples that behaved similarly to the BRAF+TERT-positive samples, indicating other second hit alterations that indicate aggressiveness, including TP53 or PIK3CA gene mutations? Do the 2 samples with TERT promoter mutations and no BRAF mutation harbor other drivers of aggressive disease such as certain gene fusions?
- Discussion line 199 states 14/724 samples in reference 7 were positive for TERT C228T mutation. Line 211 says 12 patients from reference 7 were positive for TERT C228T. Please correct.
- Discussion line 200 states 4/724 samples in reference 7 were positive for TERT C216T mutation. Line 211 says 1 was positive for C216T. Please correct.
- For 8 and 9 – if you are trying to say your results of 12 positive for C228T and 1 positive for C216T were similar to reference 7 results, please rephrase as that is not the meaning of the current sentence.
- Line 59 says goal was to evaluate TERT mutation status in “consecutive cases” but Methods section indicates testing was only performed in patients who consented (line 67). Please clarify if actual tested patients came from sequential samples.
Author Response
We sincerely appreciate your review. We have revised your findings as follows.
All changes according to “reviewer 1” are highlighted as “blue” color in the revised manuscript.
- TYPOGRAPHICAL ERROR. Introduction, line 55. There are excessive spaces and lack of commas when describing the genomic location of the TERT 228 and 250 known hotspot locations.
Ans)
Thanks for your point. We have made the following correction at your point.
(Before)
In thyroid cancer, three major TERT promoter mutations have been reported, namely, chr5:1, 295 228 C > T (C228T) and chr5:1, 295 250 C > T (C250T), which represent nucleotide changes of −124 C>T and −146 C>T from the ATG translation start codon in the promoter site of the TERT gene.
(After)
In TC, three major TERT promoter mutations have been reported, namely, chr5:1, 295 228 C>T(C228T) and chr5:1, 295 250 C>T(C250T), which represent nucleotide changes of −124 C>T and −146 C>T from the ATG translation start codon in the promoter site of the TERT gene.
- TYPOGRAPHICAL ERROR. Methods, section 2.1 Patient data mentions that “From February 2019 to December 2021…” Please correct, as Dec 2021 is in the future.
Ans)
Thanks for your important point-out and sorry for our mistake. We made the following correction.
(Before)
From February 2019 to December 2021, 622 thyroid cancer patients were tested for the BRAFV600E and TERT promoter mutations in their surgically resected thyroid cancer tissue at Inha University Hospital, Incheon, Korea.
(After)
From February 2019 to December 2020, we analyzed 622 thyroid cancer patients who were tested and reported the full result for the BRAFV600E and TERT promoter mutations in their surgically resected thyroid cancer tissue at Inha University Hospital, Incheon, Korea.
- Methods for BRAF p.V600E and TERT promoter mutation detections should be more thorough. Specifically, vendor details for the kit listed for BRAF detection shows this method by qPCR which is a very sensitive method for mutation detection. However, TERT promoter mutation analysis was done by Sanger sequencing, which has an analytical sensitivity of roughly 20%. Please elaborate on the mutation detection methods in the Methods section. Different analytical sensitives of the methods used for BRAF and TERT mutation detection may yield false negative results for one of the assays.
.
Ans)
Thank you for your opinion about the method section. As you mentioned, sensitivity of SANGER sequencing in TERT promotor site is relatively lower than that of BRAFV600E mutation detection method which is very sensitive according to qPCR. However, as of 2021, the only approved method for TERT promotor mutation in Korea is SANGER sequencing. We added the detailed description about the TERT promotor area sequencing as follow.
(Before)
TERT promoter mutation testing was performed by the Department of Pathology, Seoul St. Mary's Hospital. The TERT promoter site was amplified by polymerase chain reaction (PCR), and bidirected Sanger sequencing was applied as described elsewhere.
(After)
TERT promoter mutation testing was performed in the Department of Pathology, Seoul St. Mary's Hospital, Korea. Amplification of TERT promoter was done by nested polymerase chain reaction (PCR). Initially, 235-bp PCR amplicon was amplified using forward 5'-AGTGGATTCGCGGGCACAGA-3' and reverse 5'-CAGCGCTGCCTGAAACTC-3' primers. Second, 163-bp PCR amplicon was amplified using forward 5'-GTCCTGCCCCTTCACCTT-3' and reverse 5'-CAGCGCTGCCTGAAACTC-3' primers. Bidirectional Sanger sequencing in both directions was done using primers that used in the second PCR.
- Additionally, specific to Sanger sequencing of TERT promoter region – this is a very GC-rich region. Did Sanger sequencing work and generate a result, either positive or negative, for 100% of the 622 sample cohort?
Ans)
Your opinion is a very good point. In fact, more than 500 thyroid surgeries are performed annually in our hospital, and only patients with negative or positive TERT mutation test results were included in this study. We added minor comment in the method section as follow.
(Before)
… 622 thyroid cancer patients were tested for the BRAFV600E and TERT promoter mutations in their surgically resected thyroid cancer tissue at Inha University Hospital, Incheon, Korea.
(After)
… we analyzed 622 thyroid cancer patients who were tested and reported the full result for the BRAFV600E and TERT promoter mutations in their surgically resected thyroid cancer tissue at Inha University Hospital, Incheon, Korea.
- A meta-analysis performed by Liu et. al. (PMID 27833153) shows the average prevalence of TERT promoter mutations in PTC to be ~10%. Your study showed approx. 2% prevalence. What may contribute to very low TERT mutation positive prevalence?
Ans)
Thank you for your opinion. Perhaps the lower proportion of TERT promotor mutations reported in our study appears to be a racial trait that occurs in a cohort consisting only of Koreans. As mentioned in discussion section, proportion of TERT promotor mutation conducted by another institution in Korea was also about the 2% and our study also reported as 2%. We added some sentences about this matter in the discussion section as follow.
(Before)
According to our results, the incidence of TERT promoter mutation was 2.1% (13 of 622). This result is similar to a previous study in Korea by Kim et al. (7) in which 12 patients carried the C228T mutation and 1 patient carried the C216T mutation. C250T mutation was not identified in our cohort.
(After)
According to our results, the incidence of TERT promoter mutation was 2.1% (13 of 622), in which 12 patients carried the C228T mutation and 1 patient carried the C216T mutation. C250T mutation was not identified in our cohort. This result is similar to a previous study in Korea by Kim et al. (7). Indicence of TERT promotor mutation in Korean cohort was about 2%, which is less frequent than previous studies about 10% of TERT promotor mutation. This is probably caused by racial differences but further research is needed.
- Since the significance of TERT C216T is unknown, and is acknowledged in the manuscript as such, please describe why it was included in the group of samples with identified TERT C228T mutations. The manuscript is written such that identification of a TERT promoter mutation correlates with more aggressive PTC behavior. Since C216T mutation has unknown significance, please elaborate on inclusion of the sample with this variant with all other TERT C228T positive cases, as there is no data to support its association with aggressiveness?
Ans)
Thank you for your comment. As you have pointed out, the C216T has yet to be known for its clinical significance. However, because the same C216T mutation was found in a study conducted by another institution in Korea (Kim et al, PMID 32527075), we also included C216T and analyzed it. So far, there is no report on whether C216T is associated with aggressive factors. I think more research is needed on this. We already described about the C216T in the discussion section, and highlighted as blue text color.
- The aim of this paper was “to evaluate the frequency of TERT promoter mutations in consecutive cases of PTC and its clinical meaning…” Prevalence data and clinical features of the tumors and disease at diagnosis are presented in this paper. Please elaborate on the clinical meaning. This is where the manuscript is really lacking merit. Is there any follow up data for the 622 sample cohort? There was mention of only one follow-up in line 222. There are several mentions of the association of TERT promoter mutations with poor prognosis. Is that what was found in this patient set? Did any patients present with distant metastasis? Were there BRAF-positive, TERT-negative samples that behaved similarly to the BRAF+TERT-positive samples, indicating other second hit alterations that indicate aggressiveness, including TP53 or PIK3CA gene mutations? Do the 2 samples with TERT promoter mutations and no BRAF mutation harbor other drivers of aggressive disease such as certain gene fusions?
Ans)
Thank you for your profound comment. For the clinical meaning, we found that TERT promotor mutation is associated with aggressive behavior of PTC including lymphatic invasion, vascular invasion, BRAF V600E mutation, advanced TN stage, overall stage and higher MACIS risk score. According to it, we can suggest that harboring TERT promotor mutation is associated with aggressive feature of PTC. About the follow up, we started the TERT promotor mutation from the 2019, only 2 years. During this period, only 1 patient showed recurrence as mentioned in our manuscript. As you said about the follow up, we need more than 5-10 years of observational data to see long-term relapses or metastases for the PTC patients. In addition, a greater number of samples are needed than now to see the effect of the second hit mutations in BRAF and TERT. Only we have the 13 patients with TERT promotor mutation. Finally, to investigate the other gene mutations or fusions, it is necessary to newly plan and proceed with other studies rather than in this study. Thank you for your deep reflection.
- Discussion line 199 states 14/724 samples in reference 7 were positive for TERT C228T mutation. Line 211 says 12 patients from reference 7 were positive for TERT C228T. Please correct.
Ans)
Thanks for your point. We corrected it as follow.
In this study, the detection rate of TERT promoter mutation was 2.8% (20/724); C228T mutation was found in 1.9% (14/724), and C250T mutation was identified in 0.3% (2/724). A novel C216T mutation was identified in 0.6% (4/724).
- Discussion line 200 states 4/724 samples in reference 7 were positive for TERT C216T mutation. Line 211 says 1 was positive for C216T. Please correct.
Ans)
Thanks for your point. We corrected it as follow.
In this study, the detection rate of TERT promoter mutation was 2.8% (20/724); C228T mutation was found in 1.9% (14/724), and C250T mutation was identified in 0.3% (2/724). A novel C216T mutation was identified in 0.6% (4/724).
- For 8 and 9 – if you are trying to say your results of 12 positive for C228T and 1 positive for C216T were similar to reference 7 results, please rephrase as that is not the meaning of the current sentence.
Ans)
Thanks for your point. We have modified it according to your point of view.
(Before)
According to our results, the incidence of TERT promoter mutation was 2.1% (13 of 622). This result is similar to a previous study in Korea by Kim et al. (7) in which 12 patients carried the C228T mutation and 1 patient carried the C216T mutation. C250T mutation was not identified in our cohort. The TERT promoter mutation without the BRAF V600E mutation was detected in 2 patients (15.4%), which is also similar to previous studies reporting approximately 6.3-18.5%.
(After)
According to our results, the incidence of TERT promoter mutation was 2.1% (13 of 622), in which 12 patients carried the C228T mutation and 1 patient carried the C216T mutation. C250T mutation was not identified in our cohort. Indicence of TERT promotor mutation in Korean cohort was about 2%, which is less frequent than previous studies about 10% of TERT promotor mutation.
- Line 59 says goal was to evaluate TERT mutation status in “consecutive cases” but Methods section indicates testing was only performed in patients who consented (line 67). Please clarify if actual tested patients came from sequential samples.
Ans)
Thanks for your point. We have modified it according to your point of view.
(Before)
Our aim was therefore to evaluate the frequency of TERT promoter mutations in consecutive cases of PTC and its clinical meaning in Korean PTC patients at a single institution.
(After)
Our aim was therefore to evaluate the frequency of TERT promoter mutations of PTC and its clinical meaning in Korean PTC patients at a single institution.

Reviewer 2 Report
Yun Suk Choi and colleagues present the manuscript entitled “Prospective Analysis of TERT Promoter Mutations in Papillary Thyroid Carcinoma at a Single Institution” where they have studied the frequency and the clinical impact of TERT promoter mutation in Papillary thyroid carcinoma.
The paper is interesting and well organized even if the number of cases is limited. It requires minor changes.
MINOR COMMENTS:
1. Line 10: write “Thyroid cancers (TCs)” instead of only Thyroid cancers. In this way in the subsequent paragraphs you may use also TC instead of thyroid cancer all the times;
2. In the introduction the authors should better describe the thyroid cancer classification diagnosis and therapy besides prognosis of PTC;
3. Lane 36: write “Differentiated Thyroid Cancer (DTC)” instead of DTC;
4. Lane 40: “(PDTC)” instead of (PD);
5. Lane 50: “Telomerase reverse transcriptase (TERT)” must be utilized the first time that the term TERT is introduced;
6. Table 2: Specify the classic histological TC type “PTC” instead of classic;
7. Lane 171: write “preserve” instead of maintain.
Author Response
We sincerely appreciate your review. We have revised your findings as follows.
All changes according to “reviewer 1” are highlighted as “red” color in the revised manuscript.
- Line 10: write “Thyroid cancers (TCs)” instead of only Thyroid cancers. In this way in the subsequent paragraphs you may use also TC instead of thyroid cancer all the times;
Ans)
Thanks for your point. We corrected the term of “thyroid cancer” into “TC”.
- In the introduction the authors should better describe the thyroid cancer classification diagnosis and therapy besides prognosis of PTC;
Ans)
Thanks for your point. We corrected the part as follow.
(Before)
Despite the high incidence of thyroid cancer, most thyroid cancers are differentiated tu-mors that carry a good prognosis, and DTC is reported to account for 98.3% of all thyroid cancers. Of DTC cases, 79.3% are papillary thyroid carcinoma (PTC).
(After)
Thyroid cancers are classified as differentiated thyroid cancer (DTC) including papillary thyroid carcinoma (PTC) and follicular thyroid carcinoma (FTC); poorly differentiated thyroid carcinoma (PDTC) and anaplastic thyroid carcinoma (ATC); medullary thyroid carcinoma (MTC); and lymphoma. Among them, differentiated cancer is reported to account for 98.3% of all thyroid cancers, and 79.3% are papillary thyroid carcinoma (PTC). Treatment principle of thyroid cancer is the surgical resection of thyroid gland with regional lymph node dissection, and optional radioactive iodine therapy.
- Lane 36: write “Differentiated Thyroid Cancer (DTC)” instead of DTC;
Ans)
Thanks for your point. We have corrected the term.
(Before)
Despite the high incidence of thyroid cancer, most thyroid cancers are differentiated tumors that carry a good prognosis, and DTC is reported to account for 98.3% of all thyroid cancers. Of DTC cases, 79.3% are papillary thyroid carcinoma (PTC). (3)
(After)
Thyroid cancers are classified as differentiated thyroid cancer (DTC) including papillary thyroid carcinoma (PTC) and follicular thyroid carcinoma (FTC); poorly differentiated thyroid carcinoma (PDTC) and anaplastic thyroid carcinoma (ATC); medullary thyroid carcinoma (MTC); and lymphoma. Among them, differentiated cancer is reported to account for 98.3% of all thyroid cancers, and 79.3% are papillary thyroid carcinoma (PTC). Treatment principle of thyroid cancer is the surgical resection of thyroid gland with regional lymph node dissection, and optional radioactive iodine therapy.
- Lane 40: “(PDTC)” instead of (PD);
Ans)
Thanks for your point. We corrected the term “PD” into “PDTC”.
- Lane 50: “Telomerase reverse transcriptase (TERT)” must be utilized the first time that the term TERT is introduced;
Ans)
Thanks for your point. We described the full term of TERT.
- Table 2: Specify the classic histological TC type “PTC” instead of classic;
Ans)
Thanks for your point. We corrected the term of “classic” into “PTC (Classic type)”.
- Lane 171: write “preserve” instead of maintain.
Ans)
Thanks for your point. We corrected the term of “maintain” into “preserve”.

Round 2
Reviewer 1 Report
Thank you for allowing me to review the modified manuscript. All requested corrections or clarifications are adequate.